# Sample Complexity Bounds for Iterative Stochastic Policy Optimization

**Marin Kobilarov**
Department of Mechanical Engineering
Johns Hopkins University
Baltimore, MD 21218
marin@jhu.edu

## Abstract

This paper is concerned with robustness analysis of decision making under uncertainty. We consider a class of iterative stochastic policy optimization problems and analyze the resulting expected performance for each newly updated policy at each iteration. In particular, we employ concentration-of-measure inequalities to compute future expected cost and probability of constraint violation using empirical runs. A novel inequality bound is derived that accounts for the possibly unbounded change-of-measure likelihood ratio resulting from iterative policy adaptation. The bound serves as a high-confidence certificate for providing future performance or safety guarantees. The approach is illustrated with a simple robot control scenario and initial steps towards applications to challenging aerial vehicle navigation problems are presented.

## 1 Introduction

We consider a general class of stochastic optimization problems formulated as

$$\xi^* = \arg \min_{\xi} \mathbb{E}_{\tau \sim p(\cdot|\xi)}[J(\tau)], \tag{1}$$

where $\xi$ defines a vector of decision variables, $\tau$ represents the system response defined through the density $p(\tau|\xi)$, and $J(\tau)$ defines a positive cost function which can be non-smooth and non-convex. It is assumed that $p(\tau|\xi)$ is either known or can be sampled from, e.g. in a black-box manner. The objective is to obtain high-confidence sample complexity bounds on the expected cost for a given decision strategy by observing past realizations of possibly different strategies. Such bounds are useful for two reasons: 1) for providing robustness guarantees for future executions, and 2) for designing new algorithms that directly minimize the bound and therefore are expected to have built-in robustness.

Our primary motivation arises from applications in robotics, for instance when a robot executes control policies to achieve a given task such as navigating to a desired state while perceiving the environment and avoiding obstacles. Such problems are traditionally considered in the framework of reinforcement learning and addressed using policy search algorithms, e.g. [1, 2] (see also [3] for a comprehensive overview with a focus on robotic applications [4]). When an uncertain system model is available the problem is equivalent to robust model-predictive control (MPC) [5].

Our specific focus is on providing formal guarantees on future executions of control algorithms in terms of maximum expected cost (quantifying performance) and maximum probability of constraint violation (quantifying safety). Such bounds determine the reliability of control in the presence of process, measurement and parameter uncertainties, and contextual changes in the task. In this work we make no assumptions about nature of the system structure, such as linearity, convexity, or Gaussianity. In addition, the proposed approach applies either to a physical system without an available

model, to an analytical stochastic model, or to a white-box model (e.g. from a high-fidelity open-source physics engine). In this context, PAC bounds have been rarely considered but could prove essential for system certification, by providing high-confidence guarantees for future performance and safety, for instance "with 99% chance the robot will reach the goal within 5 minutes", or "with 99% chance the robot will not collide with obstacles".

**Approach.** To cope with such general conditions, we study robustness through a statistical learning viewpoint [6, 7, 8] using *finite-time* sample complexity bounds on performance based on empirical runs. This is accomplished using concentration-of-measure inequalities [9] which provide only probabilistic bounds , i.e. they certify the algorithm execution in terms of statements such as: "in future executions, with 99% chance the expected cost will be less than X and the probability of collision will be less than Y". While such bounds are generally applicable to any stochastic decision making process, our focus and initial evaluation is on stochastic control problems.

**Randomized methods in control analysis.** Our approach is also inspired by existing work on randomized algorithms in control theory originally motivated by robust linear control design [10]. For example, early work focused on probabilistic root-locus design [11] and later applied to constraint satisfaction [12] and general cost functions [13]. High-confidence bounds for decidability of linear stability were refined in [14]. These are closely related to the concepts of randomized stability robustness analysis (RSRA) and randomized performance robustness analysis (RPRA) [13]. Finite-time probabilistic bounds for system identification problems have also been obtained through a statistical learning viewpoint [15].

## 2 Iterative Stochastic Policy Optimization

Instead of directly searching for the optimal $\xi$ to solve (1) a common strategy in direct policy search and global optimization [16, 17, 18, 19, 20, 21] is to iteratively construct a surrogate stochastic model $\pi(\xi|\nu)$ with hyper-parameters $\nu \in \mathcal{V}$, such as a Gaussian Mixture Model (GMM), where $\mathcal{V}$ is a vector space. The model induces a joint density $p(\tau, \xi|\nu) = p(\tau|\xi)\pi(\xi|\nu)$ encoding *natural stochasticity* $p(\tau|\xi)$ and *artificial control-exploration stochasticity* $\pi(\xi|\nu)$. The problem is then to find $\nu$ to minimize the expected cost

$$\mathcal{J}(v) \triangleq \mathbb{E}_{\tau,\xi \sim p(\cdot|\nu)}[J(\tau)],$$

iteratively until convergence, which in many cases also corresponds to $\pi(\cdot|\nu)$ shrinking close to a delta function around the optimal $\xi^*$ (or to multiple peaks when multiple disparate optima exist as long as $\pi$ is multi-modal).

The typical flow of the iterative policy optimization algorithms considered in this work is:

> Iterative Stochastic Policy Optimization (ISPO)
> 0. Start with initial hyper-parameters $\nu_0$ (i.e. a prior), set $i = 0$
> 1. Sample $M$ trajectories $(\xi_j, \tau_j) \sim p(\cdot|\nu_i)$ for $j = 1, \ldots, M$
> 2. Compute new policy $\nu_{i+1}$ using observed costs $J(\tau_j)$
> 3. Compute bound on expected cost and Stop if below threshold, else set $i = i+1$ and Goto 1

The purpose of computing probably-approximate bounds is two-fold: a) to analyze the performance of such standard policy search algorithms; b) to design new algorithms by not directly minimizing an estimate of the expected cost, but by minimizing an *upper confidence bound* on the expected cost instead. The computed policy will thus have "built-in" robustness in the sense that, with high-probability, the resulting cost will not exceed an a-priori known value. The present paper develops bounds applicable to both (a) and (b), but only explores their application to (a), i.e. to the analysis of existing iterative policy search methods.

**Cost functions.** We consider two classes of cost functions $J$. The first class encodes *system performance* and is defined as a bounded real-valued function such that $0 \leq J(\tau) \leq b$ for any $\tau$. The second are binary-valued indicator functions representing *constraint violation*. Assume that the variable $\tau$ must satisfy the condition $g(\tau) \leq 0$. The cost is then defined as $J(\tau) = I_{\{g(\tau)>0\}}$ and its expectation can be regarded as the probability of constraint violation, i.e.

$$\mathbb{P}(g(\tau) > 0) = \mathbb{E}_{\tau \sim p(\cdot|\xi)} I_{\{g(\tau)>0\}}.$$

In this work, we will be obtain bounds for both classes of cost functions.

## 3 A Specific Application: Discrete-time Stochastic Control

We next illustrate the general stochastic optimization setting using a classical discrete-time non-linear optimal control problem. Specific instances of such control problems will later be used for numerical evaluation. Consider a discrete-time dynamical model with state $x_k \in X$, where $X$ is an $n$-dimensional manifold, and control inputs $u_k \in \mathbb{R}^m$ at time $t_k \in [0, T]$ where $k = 0, \ldots, N$ denotes the time stage. Assume that the system dynamics are given by

$$x_{k+1} = f_k(x_k, u_k, w_k), \qquad \text{subject to} \quad g_k(x_k, u_k) \leq 0, \ \ g_N(x_N) \leq 0,$$

where $f_k$ and $g_k$ correspond either to the physical plant, to an analytical model, or to a "white-box" high-fidelity physics-engine update step. The terms $w_k$ denotes process noise. Equivalently, such a formulation induces the *process model* density $p(x_{k+1}|x_k, u_k)$. In addition, consider the cost

$$J(x_{0:N}, u_{0:N-1}) \triangleq \sum_{k=0}^{N-1} L_k(x_k, u_k) + L_N(x_N),$$

where $x_{0:N} \triangleq \{x_0, \ldots, x_N\}$ denotes the complete trajectory and $L_k$ are given nonlinear functions. Our goal is to design feedback control policies to optimize the expected value of $J$. For simplicity, we will assume perfect measurements although this does not impose a limitation on the approach.

Assume that any decision variables in the problem (such as feedforward or feedback gains, obstacle avoidance terms, mode switching variables) are encoded using a finite-dimensional vector $\xi \in \mathbb{R}^{n_\xi}$ and define the control law $u_k = \Phi_k(x_k)\xi$ using basis functions $\Phi_k(x) \in \mathbb{R}^{m \times n_\xi}$ for all $k = 0, \ldots, N-1$. This representation captures both static feedback control laws as well as *time-varying optimal control laws* of the form $u_k = u_k^* + K_k^{LQR}(x_k - x_k^*)$ where $u_k^* = B(t_k)\xi$ is an optimized feedforward control (parametrized using basis functions $B(t) \in \mathbb{R}^{m \times z}$ such as B-splines), $K_k^{LQR}$ is the optimal feedback gain matrix of the LQR problem based on the linearized dynamics and second-order cost expansion around the optimized nominal reference trajectory $x^*$, i.e. such that $x_{k+1}^* = f_k(x_k^*, u_k^*, 0)$.

The complete trajectory of the system is denoted by the random variable $\tau = (x_{0:N}, u_{0:N-1})$ and has density $p(\tau|\xi) = p(x_0)\Pi_{k=0}^{N-1} p(x_{k+1}|x_k, u_k)\delta(u_k - \Phi_k(x_k)\xi)$, where $\delta(\cdot)$ is the Dirac delta. The trajectory constraint takes the form $\{g(\tau) \leq 0\} \triangleq \bigwedge_{k=0}^{N-1} \{g_k(x_k, u_k) \leq 0\} \wedge \{g_N(x_N) \leq 0\}$.

**A simple example.**   As an example, consider a point-mass robot modeled as a double-integrator system with state $x = (p, v)$ where $p \in \mathbb{R}^d$ denotes position and $v \in \mathbb{R}^d$ denotes velocity with $d = 2$ for planar workspaces and $d = 3$ for 3-D workspaces. The dynamics is given, for $\Delta t = T/N$, by

$$p_{k+1} = p_k + \Delta t v_k + \frac{1}{2}\Delta t^2(u_k + w_k),$$

$$v_{k+1} = v_k + \Delta t(u_k + w_k),$$

where $u_k$ are the applied controls and $w_k$ is zero-mean white noise. Imagine that the constraint $g_k(x, u) \leq 0$ defines circular obstacles $\mathcal{O} \subset \mathbb{R}^d$ and control norm bounds defined as

$$r_o - \|p - p_o\| \leq 0, \qquad \|u\| \leq u_{\max},$$

where $r_o$ is the radius of an obstacle at position $p_o \in \mathbb{R}^d$. The cost $J$ could be arbitrary but a typical choice is $L(x, u) = \frac{1}{2}\|u\|_R^2 + q(x)$ where $R > 0$ is a given matrix and $q(x)$ is a nonlinear function defining a task. The final cost could force the system towards a goal state $x_f \in \mathbb{R}^n$ (or a region $X_f \subset \mathbb{R}^n$) and could be defined according to $L_N(x) = \frac{1}{2}\|x - x_f\|_{Q_f}^2$ for some given matrix $Q_f \geq 0$. For such simple systems one can choose a smooth feedback control law $u_k = \Phi_k(x)\xi$ with static positive gains $\xi = (k_p, k_d, k_o) \in \mathbb{R}^3$ and basis function

$$\Phi(x) = [\ p_f - p \quad v_f - v \quad \varphi(x, \mathcal{O})\ ],$$

where $\varphi(x, \mathcal{O})$ is an obstacle-avoidance force, e.g. defined as the gradient of a potential field or as a gyroscopic "steering" force $\varphi(x, \mathcal{O}) = s(x, \mathcal{O}) \times v$ that effectively rotates the velocity vector [22] . Alternatively, one could employ a time-varying optimal control law as described in §3.

# 4 PAC Bounds for Iterative Policy Adaptation

We next compute probabilistic bounds on the expected cost $\mathcal{J}(\nu)$ resulting from the execution of a new stochastic policy with hyper-parameters $\nu$ using observed samples from previous policies $\nu_0, \nu_1, \dots$. The bound is agnostic to how the policy is updated (i.e. Step 2 in the ISPO algorithm).

## 4.1 A concentration-of-measure inequality for policy adaptation

The stochastic optimization setting naturally allows the use of a *prior belief* $\xi \sim \pi(\cdot|\nu_0)$ on what good control laws could be, for some known $\nu_0 \in \mathcal{V}$. After observing $M$ executions based on such prior, we wish to find a new improved policy $\pi(\cdot|\nu)$ which optimizes the cost

$$\mathcal{J}(\nu) \triangleq \mathbb{E}_{\tau,\xi \sim p(\cdot|\nu)}[J(\tau)] = \mathbb{E}_{\tau,\xi \sim p(\cdot|\nu_0)}\left[J(\tau)\frac{\pi(\xi|\nu)}{\pi(\xi|\nu_0)}\right], \tag{2}$$

which can be approximated using samples $\xi_j \sim \pi(\xi|\nu_0)$ and $\tau_j \sim p(\tau|\xi_j)$ by the empirical cost

$$\frac{1}{M}\sum_{j=1}^{M}\left[J(\tau_j)\frac{\pi(\xi_j|\nu)}{\pi(\xi_j|\nu_0)}\right]. \tag{3}$$

The goal is to compute the parameters $\nu$ using the sampled decision variables $\xi_j$ and the corresponding observed costs $J(\tau_j)$. Obtaining practical bounds for $\mathcal{J}(\nu)$ becomes challenging since the change-of-measure likelihood ratio $\frac{\pi(\xi|\nu)}{\pi(\xi|\nu_0)}$ can be *unbounded* (or have very large values) [23] and a standard bound, e.g. such as Hoeffding's or Bernstein's becomes impractical or impossible to apply. To cope with this we will employ a recently proposed *robust estimation* [24] technique stipulating that instead of estimating the expectation $m = \mathbb{E}[X]$ of a random variable $X \in [0,\infty)$ using its empirical mean $\widehat{m} = \frac{1}{M}\sum_{j=1}^{M} X_j$, a more robust estimate can be obtained by truncating its higher moments, i.e. using $\widehat{m}_\alpha \triangleq \frac{1}{\alpha M}\sum_{j=1}^{M} \psi(\alpha X_j)$ for some $\alpha > 0$ where

$$\psi(x) = \log(1 + x + \frac{1}{2}x^2). \tag{4}$$

What makes this possible is the key assumption that the (possibly unbounded) random variable must have *bounded second moment*. We employ this idea to deal with the unboundedness of the policy adaptation ratio by showing that in fact its second moment can be bounded and corresponds to an information distance between the current and previous stochastic policies.

To obtain sharp bounds though it is useful to employ samples over multiple iterations of the ISPO algorithm, i.e. from policies $\nu_0, \nu_1, \dots, \nu_{L-1}$ computed in previous iterations. To simplify notation let $z = (\tau, \xi)$ and define $\ell_i(z, \nu) \triangleq J(\tau)\frac{\pi(\xi|\nu)}{\pi(\xi|\nu_i)}$. The cost (2) of executing $\nu$ can now be equivalently expressed as

$$\mathcal{J}(\nu) \equiv \frac{1}{L}\sum_{i=0}^{L-1}\mathbb{E}_{z \sim p(\cdot|\nu_i)}\ell_i(z, \nu)$$

using the computed policies in previous iterations $i = 0, \dots, L-1$. We next state the main result:

**Proposition 1.** *With probability $1 - \delta$ the expected cost of executing a stochastic policy with parameters $\xi \sim \pi(\cdot|\nu)$ is bounded according to:*

$$\mathcal{J}(\nu) \leq \inf_{\alpha>0}\left\{\widehat{\mathcal{J}}_\alpha(\nu) + \frac{\alpha}{2L}\sum_{i=0}^{L-1}b_i^2 e^{D_2(\pi(\cdot|\nu)||\pi(\cdot|\nu_i))} + \frac{1}{\alpha LM}\log\frac{1}{\delta}\right\}, \tag{5}$$

*where $\widehat{\mathcal{J}}_\alpha(\nu)$ denotes a robust estimator defined by*

$$\widehat{\mathcal{J}}_\alpha(\nu) \triangleq \frac{1}{\alpha LM}\sum_{i=0}^{L-1}\sum_{j=1}^{M}\psi\left(\alpha\ell(z_{ij}, \nu)\right),$$

*computed after $L$ iterations, with $M$ samples $z_{i1}, \dots, z_{iM} \sim p(\cdot|\nu_i)$ obtained at iterations $i = 0, \dots, L-1$, where $D_\beta(p||q)$ denotes the Renyii divergence between $p$ and $q$ defined by*

$$D_\beta(p||q) = \frac{1}{\beta - 1}\log\int\frac{p^\beta(x)}{q^{\beta-1}(x)}dx.$$

*The constants $b_i$ are such that $0 \leq J(\tau) \leq b_i$ at each iteration $i = 0, \dots, L-1$.*

*Proof.* The bound is obtained by relating the mean to its robust estimate according to

$$
\mathbb{P}\left(LM(\mathcal{J}(\nu) - \widehat{\mathcal{J}}_\alpha(\nu)) \geq t\right)
$$
$$
= \mathbb{P}\left(e^{\alpha LM(\mathcal{J}(\nu) - \widehat{\mathcal{J}}_\alpha(\nu))} \geq e^{\alpha t}\right),
$$
$$
\leq \mathbb{E}\left[e^{\alpha LM(\mathcal{J}(\nu) - \widehat{\mathcal{J}}_\alpha(\nu))}\right] e^{-\alpha t}, \tag{6}
$$
$$
= e^{-\alpha t + \alpha LM\mathcal{J}(\nu)} \mathbb{E}\left[e^{\sum_{i=0}^{L-1}\sum_{j=1}^{M} -\psi(\alpha \ell_i(z_{ij},\nu))}\right]
$$
$$
= e^{-\alpha t + \alpha LM\mathcal{J}} \mathbb{E}\left[\prod_{i=0}^{L-1}\prod_{j=1}^{M} e^{-\psi(\alpha \ell_i(z_{ij},\nu))}\right]
$$
$$
= e^{-\alpha t + \alpha LM\mathcal{J}} \prod_{i=0}^{L-1}\prod_{j=1}^{M} \mathbb{E}_{z\sim p(\cdot|\nu_i)}\left[1 - \alpha \ell_i(z,\nu) + \frac{\alpha^2}{2}\ell_i(z,\nu)^2\right] \tag{7}
$$
$$
= e^{-\alpha t + \alpha LM\mathcal{J}(\nu)} \prod_{i=0}^{L-1}\prod_{j=1}^{M}\left(1 - \alpha\mathcal{J}(\nu) + \frac{\alpha^2}{2}\mathbb{E}_{z\sim p(\cdot|\nu_i)}[\ell_i(z,\nu)^2]\right)
$$
$$
\leq e^{-\alpha t + \alpha LM\mathcal{J}(\nu)} \prod_{i=0}^{L-1}\prod_{j=1}^{M} e^{-\alpha\mathcal{J}(\nu) + \frac{\alpha^2}{2}\mathbb{E}_{z\sim p(\cdot|\nu_i)}[\ell_i(z,\nu)^2]} \tag{8}
$$
$$
\leq e^{-\alpha t + M\frac{\alpha^2}{2}\sum_{i=0}^{L-1}\mathbb{E}_{z\sim p(\cdot|\nu_i)}[\ell_i(z,\nu)^2]},
$$

using Markov's inequality to obtain (6), the identities $\psi(x) \geq -\log(1 - x + \frac{1}{2}x^2)$ in (7) and $1 + x \leq e^x$ in (8). Here, we adapted the moment-truncation technique proposed by Catoni [24] for general unbounded losses to our policy adaptation setting in order to handle the possibly unbounded likelihood ratio. These results are then combined with

$$
\mathbb{E}\left[\ell_i(z,\nu)^2\right] \leq b_i^2 \mathbb{E}_{\pi(\cdot|\nu_i)}\left[\frac{\pi(\xi|\nu)^2}{\pi(\xi|\nu_i)^2}\right] = b_i^2 e^{D_2(\pi||\pi_i)},
$$

where the relationship between the likelihood ratio variance and the Renyii divergence was established in [23]. □

Note that the Renyii divergence can be regarded as a distance between two distribution and can be computed in closed bounded form for various distributions such as the exponential families; it is also closely related to the Kullback-Liebler (KL) divergence, i.e. $D_1(p||q) = KL(p||q)$.

## 4.2 Illustration using simple robot navigation

We next illustrate the application of these bounds using the simple scenario introduced in §3. The stochasticity is modeled using a Gaussian density on the initial state $p(x_0)$, on the disturbances $w_k$ and on the goal state $x_f$. Iterative policy optimization is performed using a stochastic model $\pi(\xi|\nu)$ encoding a multivariate Gaussian, i.e.

$$
\pi(\xi|\nu) = \mathcal{N}(\xi|\mu, \Sigma)
$$

which is updated through reward-weighted-regression (RWR) [3], i.e. in Step 2 of the ISPO algorithm we take $M$ samples, observe their costs, and update the parameters according to

$$
\mu = \sum_{j=1}^{M}\bar{w}(\tau_j)\xi_j, \qquad \Sigma = \sum_{j=1}^{M}\bar{w}(\tau_j)(\xi_j - \mu)(\xi_j - \mu)^T, \tag{9}
$$

using the *tilting* weights $w(\tau) = e^{-\beta J(\tau)}$ for some adaptively chosen $\beta > 0$ and where $\bar{w}(\tau_j) \triangleq w(\tau_j)/\sum_{\ell=1}^{M} w(\tau_\ell)$ are the normalized weights.

At each iteration $i$ one can compute a bound on the expected cost using the previously computed $\nu_0, \ldots, \nu_{i-1}$. We have computed such bounds using (5) for both the expected cost and probability of

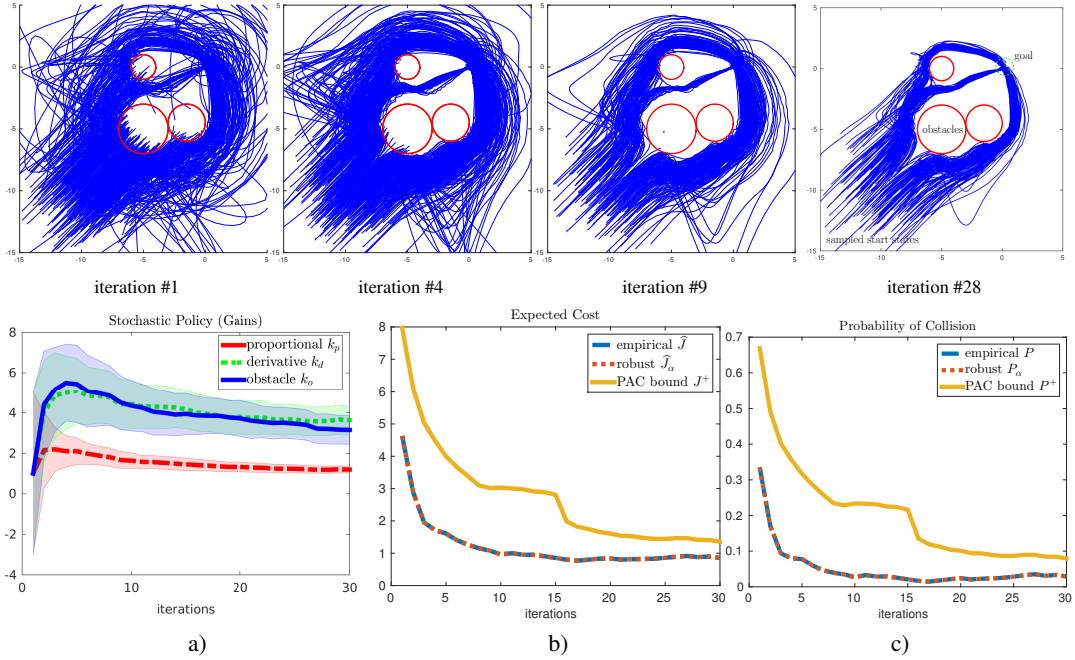

Figure 1: Robot navigation scenario based on iterative policy improvement and resulting predicted performance: a) evolution of the density $p(\xi|\nu)$ over the decision variables (in this case the control gains); b) cost function $\mathcal{J}$ and its computed upper bound $\mathcal{J}^+$ for future executions; c) analogous bounds on probability-of-collision $P$; snapshots of sampled trajectories (top). Note that the initial policy results in $\approx 30\%$ collisions. Surprisingly, the standard empirical and robust estimates are nearly identical.

collision, denoted respectively by $\mathcal{J}^+$ and $P^+$ using $M = 200$ samples (Figure 1) at each iteration. We used a window of maximum $L = 10$ previous iterations to compute the bounds, i.e. to compute $\nu_{i+1}$ all samples from densities $\nu_{i-L+1}, \nu_{i-L+2}, \ldots, \nu_i$ were used. Remarkably, using our robust statistics approach the resulting bound eventually becomes close to the standard empirical estimate $\widehat{\mathcal{J}}$. The collision probability bound $P^+$ decreses to less than $10\%$ which could be further improved by employing more samples and more iterations. The significance of these bounds is that one can stop the optimization (regarded as training) at any time and be able to predict expected performance in future executions using the newly updated policy before actually executing the policy, i.e. using the samples from the previous iteration.

Finally, the Renyii divergence term used in these computations takes the simple form

$$D_\beta \left( \mathcal{N}(\cdot|\mu_0, \Sigma_0) \| \mathcal{N}(\cdot|\mu_1, \Sigma_1) \right) = \frac{\beta}{2} \|\mu_1 - \mu_0\|^2_{\Sigma_\beta^{-1}} + \frac{1}{2(1-\beta)} \log \frac{|\Sigma_\beta|}{|\Sigma_0|^{1-\beta}|\Sigma_1|^\beta},$$

where $\Sigma_\beta = (1-\beta)\Sigma_0 + \beta\Sigma_1$.

## 4.3 Policy Optimization Methods

We do not impose any restrictions on the specific method used for optimizing the policy $\pi(\xi|\nu)$. When complex constraints are present such computation will involve a global motion planning step combined with local feedback control laws (we show such an example in §5). The approach can be used to either analyze such policies computed using any method of choice or to derive new algorithms based on minimizing the right-hand side of the bound. The method also applies to model-free learning. For instance, related to recent methods in robotics one could use reward-weighted-regression (RWR) or policy learning by weighted samples with returns (PoWeR) [3], stochastic optimization methods such as [25, 26], or using the related cross-entropy optimization [16, 27].

# 5 Application to Aerial Vehicle Navigation

Consider an aerial vehicle such as a quadrotor navigating at high speed through a cluttered environment. We are interested in minimizing a cost metric related to the total time taken and control effort required to reach a desired goal state, while maintaining low probability of collision. We employ an experimentally identified model of an AscTec quadrotor (Figure 2) with 12-dimensional state space $X = SE(3) \times \mathbb{R}^6$ with state $x = (p, R, \dot{p}, \omega)$ where $p \in \mathbb{R}^3$ is the position, $R \in SO(3)$ is the rotation matrix, and $\omega \in \mathbb{R}^3$ is the body-fixed angular velocity. The vehicle is controlled with inputs $u = (F, M) \in \mathbb{R}^4$ including the lift force $F \geq 0$ and torque moments $M \in \mathbb{R}^3$. The dynamics is

$$m\ddot{p} = Re_3 F + mg + \delta(p, \dot{p}), \tag{10}$$

$$\dot{R} = R\widehat{\omega}, \tag{11}$$

$$\mathbb{J}\dot{\omega} = \mathbb{J}\omega \times \omega + M, \tag{12}$$

where $m$ is the mass, $\mathbb{J}$–the inertia tensor, $e_3 = (0, 0, 1)$ and the matrix $\widehat{\omega}$ is such that $\widehat{\omega}\eta = \omega \times \eta$ for any $\eta \in \mathbb{R}^3$. The system is subject to initial localization errors and also to random disturbances, e.g. due to wind gusts and wall effects, defined as stochastic forces $\delta(p, \dot{p}) \in \mathbb{R}^3$. Each component in $\delta$ is zero-mean and has standard deviation of 3 Newtons, for a vehicle with mass $m \approx 1$ kg.

The objective is to navigate through a given urban environment at high speed to a desired goal state. We employ a two-stage approach consisting of an A*-based global planner which produces a sequence of local sub-goals that the vehicle must pass through. A standard nonlinear feedback backstepping controller based on a "slow" position control loop and a "fast" attitude control is employed [28, 29] for local control. In addition, and obstacle avoidance controller is added to avoid collisions since the vehicle is not expected to exactly follow the A* path. At each iteration $M = 200$ samples are taken with $1 - \delta = 0.95$ confidence level. A window of $L = 5$ past iterations were used for the bounds. The control density $\pi(\xi|\nu)$ is a single Gaussian as specified in §4.2. The most sensitive gains in the controller are the position proporitional and derivative terms, and the obstacle gains, denoted by $k_p$, $k_d$, and $k_o$, which we examine in the following scenarios:

a) *fixed goal, wind gusts disturbances, virtual environment:* the system is first tested in a cluttered simulated environment (Figure 2). The simulated vehicle travels at an average velocity of 20 m/s (see video in Supplement) and initially experiences more than 50% collisions. After a few iterations the total cost stabilizes and the probability of collision reduces to around 15%. The bound is close to the empirical estimate which indicates that it can be tight if more samples are taken. The collision probability bound is still too high to be practical but our goal was only to illustrate the bound behavior. It is also likely that our chosen control strategy is in fact not suitable for high-speed traversal of such tight environments.

b) *sparser campus-like environment, randomly sampled goals:* a more general evaluation was performed by adding the goal location to the stochastic problem parameters so that the bound will apply to any future desired goal in that environment (Figure 3). The algorithm converges to similar values as before, but this time the collision probability is smaller due to more expansive environment. In both cases, the bounds could be reduced further by employing more than $M = 200$ samples or by reusing more samples from previous runs according to Proposition 1.

# 6 Conclusion

This paper considered stochastic decision problems and focused on a probably-approximate bounds on robustness of the computed decision variables. We showed how to derive bounds for fixed policies in order to predict future performance and/or constraint violation. These results could then be employed for obtaining generalization PAC bounds, e.g. through a PAC-Bayesian approach which could be consistent with the proposed notion of policy priors and policy adaptation. Future work will develop concrete algorithms by directly optimizing such PAC bounds, which are expected to have built-in robustness properties.

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
