[Supplementary Material]

# 6 Supplement

**Proposition 2.** *With probability $1 - \delta$ the expected cost of executing a stochastic policy with parameters $\xi \sim \pi(\cdot|\nu)$ is bounded according to:*

$$\mathcal{J}(\nu) \le \inf_{\alpha > 0} \left\{ \widehat{\mathcal{J}}_\alpha(\nu) + \frac{\alpha}{2L} \sum_{i=0}^{L-1} b_i^2 e^{D_2(\pi(\cdot|\nu)||\pi(\cdot|\nu_i))} + \frac{1}{\alpha LM} \log \frac{1}{\delta} \right\}, \tag{14}$$

*where $\widehat{\mathcal{J}}_\alpha(\nu)$ denotes a robust estimator defined by*

$$\widehat{\mathcal{J}}_\alpha(\nu) \triangleq \frac{1}{\alpha L} \sum_{i=0}^{L-1} \frac{1}{M} \sum_{j=1}^{M} \psi\left(\alpha \ell_i(z_j, \nu)\right),$$

*computed after $L$ iterations, with $M$ samples $z_1, \ldots, z_M \sim p(\cdot|\nu_i)$ obtained at every iteration $i = 0, \ldots, L - 1$, and where*

$$\psi(x) = \log\left(1 + x + \frac{1}{2}x^2\right),$$

*while $D_\beta(p||q)$ denotes the Renyii divergence between $p$ and $q$ defined by*

$$D_\beta(p||q) = \frac{1}{\beta - 1} \log \int \frac{p^\beta(x)}{q^{\beta - 1}(x)} dx.$$

*The constants $b_i$ are such that $J(\tau) \le b_i$ at each iteration $i = 0, \ldots, L - 1$.*

*Proof.* Let $z = (\tau, \xi)$ and define $\ell_i(z, \nu) = J(\tau) \frac{\pi(\xi|\nu)}{\pi(\xi|\nu_i)}$. The expected value can be equivalently expressed as

$$\mathcal{J}(\nu) \equiv \frac{1}{L} \sum_{i=0}^{L-1} \mathbb{E}_{z \sim p(\cdot|\nu_i)} \ell_i(z, \nu)$$

where $\nu_i$ are the computed hyperparamters at each iteration $i = 0, \ldots, L - 1$. The bound is obtained by relating the mean to its robust estimate acoording to

$$\mathbb{P}\left(LM(\mathcal{J}(\nu) - \widehat{\mathcal{J}}_\alpha(\nu)) \ge t\right)$$

$$= \mathbb{P}\left(e^{\alpha LM(\mathcal{J}(\nu) - \widehat{\mathcal{J}}_\alpha(\nu))} \ge e^{\alpha t}\right),$$

$$\le \mathbb{E}\left[e^{\alpha LM(\mathcal{J}(\nu) - \widehat{\mathcal{J}}_\alpha(\nu))}\right] e^{-\alpha t}, \tag{15}$$

$$= e^{-\alpha t + \alpha LM \mathcal{J}(\nu)} \mathbb{E}\left[e^{\sum_{i=0}^{L-1} \sum_{j=1}^{M} -\psi\left(\alpha \ell_i(z_j, \nu)\right)}\right]$$

$$= e^{-\alpha t + \alpha LM \mathcal{J}} \mathbb{E}\left[\prod_{i=0}^{L-1} \prod_{j=1}^{M} e^{-\psi\left(\alpha \ell_i(z_j, \nu)\right)}\right]$$

$$= e^{-\alpha t + \alpha LM \mathcal{J}} \prod_{i=0}^{L-1} \prod_{j=1}^{M} \mathbb{E}_{z \sim p(\cdot|\nu_i)} \left[1 - \alpha \ell_i(z, \nu) + \frac{\alpha^2}{2} \ell_i(z, \nu)^2\right] \tag{16}$$

$$= e^{-\alpha t + \alpha LM \mathcal{J}(\nu)} \prod_{i=0}^{L-1} \prod_{j=1}^{M} \left(1 - \alpha \mathcal{J}(\nu) + \frac{\alpha^2}{2} \mathbb{E}_{z \sim p(\cdot|\nu_i)}[\ell_i(z, \nu)^2]\right)$$

$$\le e^{-\alpha t + \alpha LM \mathcal{J}(\nu)} \prod_{i=0}^{L-1} \prod_{j=1}^{M} e^{-\alpha \mathcal{J}(\nu) + \frac{\alpha^2}{2} \mathbb{E}_{z \sim p(\cdot|\nu_i)}[\ell_i(z, \nu)^2]} \tag{17}$$

$$\le e^{-\alpha t + M \frac{\alpha^2}{2} \sum_{i=0}^{L-1} \mathbb{E}_{z \sim p(\cdot|\nu_i)}[\ell_i(z, \nu)^2]},$$

using Markov's inequality to obtain (15), the identities $\psi(x) \ge -\log(1 - x + \frac{1}{2}x^2)$ in (16) and $1 + x \le e^x$ in (17). The key step to handle the possibly unbounded ratio and obtain a practical bound was the use of the robust transformation $\psi(\cdot)$ as proposed by Catoni [25]. These results are then combined with

$$\mathbb{E}\left[\ell_i(z, \nu)^2\right] \le b_i^2 \mathbb{E}_{\pi(\cdot|\nu_i)}\left[\frac{\pi(\xi|\nu)^2}{\pi(\xi|\nu_i)^2}\right] = b_i^2 e^{D_2(\pi||\pi_i)},$$

where the relationship between the likelihood ratio variance and the Renyii divergence was established in [24]. $\square$