[Reviews · NeurIPS 2015]

Submitted by Assigned_Reviewer_1

Pros: 1) Analyzing iterative stochastic policy optimization is an interesting problem. 2) The experiments are reasonably fine.

Cons: 1) The current analysis is limited to providing an upper bound for the policy evaluation. The paper does not provide an algorithm that actually uses this upper bound. The analysis is half-complete as it does not consider how the combination of both steps should work together.

2) The analysis is more or less standard and is not a significant theoretical contribution on its own. 3) The tightness of the oracle-like upper bound is not very clear. Is \hat{J}_alpha small enough, for any choice of alpha, that the upper bound based on it is meaningful?
Summary: See comments.

Submitted by Assigned_Reviewer_2

The authors derive a theorem that can be used to evaluate with high probability a bound on either the cost of a particular policy, or the degree to which it will violate given constraints. The authors suggest that this might serve as a useful way to determine convergence of policy search methods, and present experiments on simulated robot navigation using RWR as the policy optimization methods.

Overall, I found the idea of deriving such a bound to be quite intriguing, and the experimental results seem moderately realistic and quite convincing. I think this work has potential for interesting followup and will be interesting to many people.
Summary: An interesting analysis of policy search that shows bounds on cost or on violation of specific constraints, providing a practical way for determining convergence. Beyond the theoretical elegance, this idea could be quite useful in practice, as demonstrated by the experiments.

Submitted by Assigned_Reviewer_3

This paper proves PAC bounds for control policies, and gives a learning algorithm for optimizing policies for these bounds.

This is an interesting and useful problem in robotics, as the authors are correct in stating that many robotics tasks have severe penalties for "worst-case" performance, as in the example given of UAV obstacle avoidance.

Overall the mathematical formalism and readability of this paper are very good. The experiments are also interesting, as the authors connect largely theoretical work to (simulated) real-world problems.

My main concern with this paper is the lack of comparative evaluation vs. other approaches. For example, the authors cite obstacle avoidance rates improving significantly with a few iterations of their algorithm for the UAV case. It would be interesting to see how that rate compared to a more standard policy learning approach optimizing, for example, for expected reward rather than a probabilistic bound. Comparing mean performance for some metric would also be interesting, as I expect there's a tradeoff. This would significantly strengthen the paper by providing direct evidence for the authors' claims that optimizing PAC bounds helps to avoid cases like collisions.

Overall I think this is a good paper, but it could be much better with some additional evaluation as mentioned above.
Summary: This paper is theoretically very strong and attacks an interesting problem for robotics and other control applications. More extensive evaluation as compared to existing methods would significantly strengthen it.

Submitted by Assigned_Reviewer_4

- \usepackage{cite}

- l. 062: ", , i.e." - The video should be cropped and some annotations added.

Rebuttal ======== Thank you for the additional information.
Summary: LIGHT REVIEW The topic and results are highly relevant, I enjoyed the running example, and the final experiment nicely complements the paper.

Submitted by Assigned_Reviewer_5

Paper considers iterative stochastic policy optimization, and independent of how the policy is updated, the authors provide a bound on the expected cost resulting from the execution of a new stochastic policy. using observed samples from a previous policy. The bound is illustrated in an example of aerial vehicle navigation.

Quality,

Results suggest that original policy could be improved to achieve only a 10% predicted collision rate (down from 50%) - while that shows good improvement, it would be helpful to know how the original policy was chosen, as a 50% collision probability seems very high. I.e. in this case, could the authors provide evidence that a good selection of hand picked control parameters cannot do better than 50% coll prob to start?

How much does the choice of alpha in Line 245 impact the tightness of the bound? While closed form, that does not appear to be the best choice.

Clarity,

Section 2.1 seems quite separate from the rest of the paper, and given the level of specificity given in the discrete time dynamics, the concepts are not well connected to what follows. Line 105 talks about "an equivalent stochastic optimization" - are these problems really equivalent? If so, equivalent in what sense, and if not, why discuss the discrete form?

Also, the stochastic problem

on Lines 107 and 110 appear similar to chance constrained control (an MPC) - thus the authors should discuss those similarities. Numerous papers are available - the authors could start here (though many newer results exist):

@article{schwarm1999chance, title={Chance-constrained model predictive control},

author={Schwarm, Alexander T and Nikolaou, Michael},

journal={AIChE Journal}, volume={45},

number={8},

pages={1743--1752},

year={1999} }

The figure text is far too small to be readable

Legends of figs 2(b), 2(c), 3(b), 3(c) have 3 lines, but the blue one is not observable

The discussion of the results at the end of section 4 could be greatly improved - there is very little discussion of the plots in the following pages, and what is there doesn't shed much more light on the results. The graphs suggest that something interesting is going on, but exactly what is hard to know.

Originality

Authors point out that obtaining bounds for the cost is challenging because of the unbounded change-of-measure likelihood ratio, making approaches based on hoeffdings inequality impractical. They propose an alternative from Catoni [25], but that leads to statements like "The key step is to use the the [sic] moment truncation map \psi proposed by Catoni [25]..." [Line 232]. It is good to build on prior work, but this statement somewhat reduces the apparent originality of this work as it is indicates that Catoni's contribution was significant.

Significance

The result is significant in that the bounds provided will enable the development of algorithms that yield policies that are robust in the sense that the resulting cost will not exceed an apriori known cost (at least with high probability).

Minor points:

[7] seems like an odd choice to reference for robust MPC - there better papers to consider, e.g. http://homes.esat.kuleuven.be/~maapc/static/files/CACSD/survey-robust-mpc.pdf

Summary: Authors tackle an interesting problem with an algorithm that appears to being well. Improving the presentation of the figures and beefing up the discussion of the results would greatly improve the utility of the paper

Submitted by Assigned_Reviewer_6

Sample Complexity Bounds for Iterative Stochastic Policy Optimization presents a method to provide bounds for the excution cost of a policy given the excution costs of a previous policy. The bounds are computed and compared to the empirical estimate for a simple robot domain and an arial robot domain. The bounds were found for estimated costs of a policy and the probability of a crash and these bounds were found to be close to the emperical results.

The quality of the paper was nice, and the explanations clear. It would have been nice to see the effect of changing starting policies for the convergence of expected costs. The work seems original. Testing on more standard problems like grid world or cart pole would have been more helpful to compare against. The work would be significant if PAC like bounds can be directly achieved.

Summary: This work looks at computing bounds on robustness of descision making specifically for trajectory learning. This allows to compare performance of policy learning algorithms or the policy currently learned with respect to costs of the policy. The idea seems important for domains where emperical testing of policies can be expensive.

Author Feedback
Author rebuttal: We appreciate the constructive and insightful comments by the reviewers and thank them for their effort.

One of the main issues (raised by Reviewers 1 and 4) was that the paper does not provide a comparison of algorithms which use the bound vs. standard algorithms (e.g. using the empirical cost). Our goal for this paper was only to develop the bound as a ``certification'' measure (applicable to any policy algorithm) and illustrate it with examples. Nevertheless, we could easily include a comparison (using the UAV model) in the revised paper and will do so. A detailed comparison on different complex systems and convergence analysis of the complete algorithm (based on the bound) would require further analysis beyond this conference paper and will be subject of future work.

The choice of alpha, raised by Reviewer#2, is important and we have found that the optimal value is typically close to the default (closed-form) value we specified in the paper. In the revision, we could include a graph showing the actual optimized value vs. the closed-form value. We will better connect Sec 2.1 to the rest of the paper (as Reviewer#2 suggests it is currently somewhat disjoint)-- the main purpose of 2.1 was to give a motivating setting which is later illustrated by the robot navigation scenarios. We also thank for the additional references and suggestions for improving the figures which we will incorporate in the revision.

The choice of start policy (in response to Reviews#2 and 3) was such that it results in a nominally stable flight, i.e. in the absence of disturbances and assuming convex obstacles. In particular, we always set the mean of the initial gains to 1 so there is no hand-tuning. Our goal was then to show that complex obstacles and large disturbances result in many collisions and require policy adaptation. This will be clarified in the revision.

Regarding the last comment by reviewer#4: the proposition clearly shows that a bound is possible, and the empirical tests suggest that it can be tight (which also depends on ensuring that $\hat{J}_alpha$ can be made small--this is expected due to its smoothing effect, and is illustrated in the simulated scenarios).

We thank reviewers 5 and 6 for their encouraging comments and will incorporate the suggestions in Review#6 in the revision.